# PMMA/SWCNT Composites with Very Low Electrical Percolation Threshold by Direct Incorporation and Masterbatch Dilution and Characterization of Electrical and Thermoelectrical Properties

**DOI:** 10.3390/nano13081431

**Published:** 2023-04-21

**Authors:** Ezgi Uçar, Mustafa Dogu, Elcin Demirhan, Beate Krause

**Affiliations:** 1Leibniz-Institut für Polymerforschung Dresden e.V. (IPF), Hohe Str. 6, 01069 Dresden, Germany; ezgiucar@gmail.com; 2Chemical Engineering Department, Yildiz Technical University, Davutpasa Campus, Esenler, 34220 Istanbul, Türkiye; demirhan@yildiz.edu.tr; 3Mir Ar-Ge Inc., Research Department, Esenyurt, 34522 Istanbul, Türkiye; mus.dogu@gmail.com

**Keywords:** carbon nanotubes, melt-mixing, polymer composites, SWCNT, PMMA

## Abstract

In the present study, Poly(methyl methacrylate) (PMMA)/single-walled carbon nanotubes (SWCNT) composites were prepared by melt mixing to achieve suitable SWCNT dispersion and distribution and low electrical resistivity, whereby the SWCNT direct incorporation method was compared with masterbatch dilution. An electrical percolation threshold of 0.05–0.075 wt% was found, the lowest threshold value for melt-mixed PMMA/SWCNT composites reported so far. The influence of rotation speed and method of SWCNT incorporation into the PMMA matrix on the electrical properties and the SWCNT macro dispersion was investigated. It was found that increasing rotation speed improved macro dispersion and electrical conductivity. The results showed that electrically conductive composites with a low percolation threshold could be prepared by direct incorporation using high rotation speed. The masterbatch approach leads to higher resistivity values compared to the direct incorporation of SWCNTs. In addition, the thermal behavior and thermoelectric properties of PMMA/SWCNT composites were studied. The Seebeck coefficients vary from 35.8 µV/K to 53.4 µV/K for composites up to 5 wt% SWCNT.

## 1. Introduction

Poly(methyl methacrylate) (PMMA) is a transparent, odorless polymer that stands out for its mechanical properties as well as optical properties [1]. It is also chemical and atmospheric corrosion resistant. With all these properties, PMMA is also a promising polymer for sensors, optics, coating and polishing materials, binders, high voltage, and electronic applications [1,2]. PMMA is also a very suitable candidate for passive electronic components and electromagnetic interference (EMI) materials if made electrically conductive [3]. Due to its compatibility with fillers and ability to be processed easily, it is used in polymer nanocomposites by being reinforced with carbon black, graphite, metal powder, and other inorganic components [1].

Carbon-based materials have outstanding properties and can be used in a wide variety of fields with their high surface area, unique carbon structures, and ability to form covalent bonds with different elements [4,5,6,7,8,9]. They can be used directly as well as fillers in polymer composites. The interest in polymer nanocomposites reinforced with carbon-based nanofiller is increasing due to their superior properties and their lightweight. Since carbon nanotubes (CNTs) have a high aspect ratio and unique mechanical properties, they stand out as fillers in polymer nanocomposites. CNTs are hydrophobic, electrically conductive, and have a large surface area compared to other carbon-based fillers such as graphite and fullerene [10]. In particular, single-walled carbon nanotubes (SWCNTs) have unique electrical and thermal conductivity, mechanical strength, and flexibility [11]. In view of all these properties, SWCNT and SWCNT-filled composites are potential materials for electronic applicatios such as supercapacitors, sensors, and EMI materials [12,13,14,15,16]. SWCNTs differ from other CNTs in terms of the number and diameter of concentric graphene layers. SWCNTs have a low diameter (0.5–2 nm), and thus a higher aspect ratio than multi-walled carbon nanotubes (MWCNTs), which can lead to better electrical conductivity at lower loading in SWCNT composites, although an improvement in electrical conductivity depends on multiple parameters. In other words, SWCNTs can be a good choice in cases where a low percolation threshold is desired [15,17].

On the other hand, SWCNTs tend to re-aggregate easily due to van der Waals interactions between nanotubes, which complicates their dispersion in the polymer matrix [18]. In addition, SWCNTs are more difficult to disperse in the matrix because they are usually more likely to appear in bundles than MWCNTs [17]. It is known that the formation of the electrical path in the polymer matrix depends on the nanotube dispersion and distribution. The more individual SWCNTs in the matrix, the better the conductive network is formed. Thus, higher electrical conductivity and a lower electrical percolation threshold are obtained [19]. Therefore, it is crucial to provide an optimum dispersion in improving the properties of polymer composites. Solvent-based methods [18,20,21,22,23,24,25,26] with surfactants and/or ultrasonication treatment, in situ polymerization [27,28], and functionalization or modification of SWCNT [18,22,23,26,29,30] are commonly used approaches to achieve the desired distribution of SWCNT in the polymer matrix [17]. 

The melt-mixing method is preferred in the preparation of polymer composites for large-scale industrial applications [31,32]. It does not cause pollution and environmental problems since no solvent is used in the melt mixing method [32]. Therefore, preparing polymer composites with improved properties using the melt-mixing method is one of the essential issues. The manufacturing conditions play a significant role [31,33,34,35,36,37]. For example, higher temperatures can reduce the melt viscosity and decrease the shear forces. In addition, a higher rotational speed also leads to higher shear forces, which is important for the dispersion and distribution of the fillers [31,35].

Another fact is that the filler type is also decisive for the final manufacturing conditions, e.g., the feeding position during melt extrusion, as described by Müller et al. [37]. MWCNTs that require high shear stresses for their dispersion (e.g., Baytubes^®^ C150P) should be dosed in the main hopper, while for CNT powders that disperse well (e.g., NC7000), dosing into the melt at the side feeder is more advantageous.

There are limited studies on preparing PMMA/SWCNT composites with low percolation thresholds and high conductivity using melt-mixing as a simple approach. Fraser et al. [29] prepared PMMA by in situ polymerization in the presence of raw and purified SWCNTs. Then, they diluted this masterbatch with PMMA in a twin-screw extruder. Although this study stated that composites could be produced on a large scale, a conductivity value of only 1–5 × 10^−10^ S/cm was obtained (0.09 wt% loading, both SWCNT types), which was below the electrical percolation threshold [29].

There are studies in the literature where PMMA is blended with different polymers to improve the nanofiller dispersion [30,38,39]. Bikshamaiah et al. [30] incorporated functionalized SWCNTs into polyamide 6 (PA6)/PMMA blends prepared with different polymer ratios. They focused in their study on the morphological and mechanical characteristics of the composites. The tensile modulus of the blends and nanocomposites increased by ~40% and ~54%, respectively, as the PMMA ratio increased from 20 up to 80 wt% for blends and from 19.5 up to 79.5 wt% for nanocomposites. They reported that the PA6/PMMA blends exhibited two different morphologies, namely dispersed PMMA particles or co-continuous structures. In another study [39], MWCNTs were dispersed in PMMA and a PMMA/poly(vinylidene fluoride) (PVDF) blend using the melt-mixing method. It was demonstrated that PVDF improved the interactions between PMMA and MWCNTs by showing a compatibilizing effect. The electrical percolation threshold was between 0.5 and 1 wt% for both PMMA-CNT and PMMA/PVDF-CNT blend composites [39]. Guo et al. [38] dispersed CNTs in PMMA and PMMA/polystyrene (PS) blends using a twin screw micro-compounder. The percolation thresholds were 1.25 wt% and 0.5 wt% in PMMA and PMMA/PS blends, respectively.

The masterbatch dilution is another method to ensure a good dispersion and homogeneous distribution of CNTs in a polymer matrix. In this method, a composite containing a high amount of filler material is prepared, and then this composite is diluted by a second mixing process with pure polymer. The masterbatch method is known to be a simple and effective method for incorporating CNTs into the polymer matrix [40]. Pötschke et al. [41] reported that the masterbatch approach is suitable for the dispersion of MWCNTs Nanocyl^TM^ NC7000 in a polypropylene (PP) matrix and leads to a better nanotube dispersion, albeit at the expense of slightly increased electrical resistivity, compared to the MWCNT direct incorporation. Annala et al. [42] incorporated MWCNTs Nanocyl^TM^ NC7000 into PS or PMMA matrix using a masterbatch prepared by direct and in situ polymerization. In PMMA/MWCNT composites prepared by direct incorporation, the samples were not conductive up to 4 wt% CNT loading. The electrical percolation threshold of the composites using the PMMA masterbatch was below 4 wt% [42].

Studies on melt-mixed composites of thermoplastics with SWCNTs have partially focused direct incorporation method [30,43,44,45]. To the best of our knowledge, there is no study in which the methods of SWCNT addition are compared and the parameters in the preparation of the masterbatch and their effects on the properties of SWCNT nanocomposites have been examined in detail.

In addition, thermoelectric investigations of the PMMA/SWCNT composites are also of interest. Previous studies have shown that various polymer-CNT composites can be used as thermoelectric materials. If commercial CNTs, which typically show p-type behavior, are incorporated into polymers, the composites usually show the p-type as well. This could be described for melt-mixed composites based onpolypropylene (PP) [19,45,46], polycarbonate (PC) [44,47,48,49], poly(ether ether ketone) (PEEK) [49,50], PVDF) [44,51], and poly(butylene terephthalate) (PBT) [44].

In this study, PMMA/SWCNT composites were prepared by direct incorporation and masterbatch dilution using the melt mixing method. Tuball™ material, known as “graphene nanotubes”, from OCSiAl company, was used as a filler in the study. Tuball™ is a high-quality, low-cost material suitable for mass production [52]. Electrical, thermoelectrical, and morphological characterization of PMMA/SWCNT composites prepared by the direct incorporation method was performed. The effects of the rotation speed of direct incorporation and masterbatch preparation and dilution on the electrical and morphological properties of the composites were compared. 

## 2. Materials and Methods

### 2.1. Materials

Pure PMMA (Plexiglas 8N Röhm GmbH, Darmstadt, Germany) and SWCNTs Tuball^TM^ (OCSiAl S.a.r.l., Leudelange, Luxembourg) were used as polymer matrix and nanofiller, respectively. Tuball^TM^ nanotubes has a carbon purity of 75%, an outside diameter of 1.6 nm, and lengths of more than 5 μm [17,52]. The density and melt volume-flow rate values of PMMA are 1190 kg/m^3^ and 3 cm^3^/10 min (230 °C and 3.8 kg), respectively. The SWCNT selection was based on a former study comparing different kinds of CNTs [44]. The Raman spectra of this SWCNT type are published in the Appendix A [49]. The thermoelectric parameters of the SWCNT powder were described in [44]. This kind of SWCNTs is abbreviated as Tuball.

### 2.2. Composite Preparation

The melt-mixing of PMMA/SWCNT composites containing 0, 0.05, 0.075, 0.1, 0.2, 0.25, 0.5, 1, 3, and 5 wt% SWCNTs was performed in a 15 cm^3^ conical twin-screw micro compounder Xplore 15 (Xplore Instruments BV, Sittard, The Netherlands) at 260 °C for 5 min with rotation speed of 250 rpm. Processing temperature and mixing time were kept constant throughout the entire study. Before melt mixing, PMMA and SWCNTs were dried overnight in a vacuum oven at 80 °C and 120 °C, respectively.

In order to examine the effect of rotation speed at the same processing temperature and mixing time (260 °C and 5 min), composites containing 0.1, 0.25, and 1 wt% SWCNTs were prepared by direct incorporation at rotation speeds varying between 50 and 250 rpm. 

The masterbatch dilution method was used to study the effect of the nanofiller addition method on the electrical properties and nanotube macrodispersion. First, masterbatches with 5 wt% SWCNT content were prepared at 250 or 50 rpm rotation speed with the same procedure as specified in Section 2.2. Then, the masterbatches were diluted with PMMA at 250 and 50 rpm to prepare composites containing 0.05–1 wt% SWCNTs. Samples prepared by direct incorporation (direct) and masterbatch preparation (MB) and dilution (D) were named according to the corresponding rotation speeds. For example, the code of the sample prepared at 250 rpm and diluted at 50 rpm is PMMA/Tuball-MB250 + D50.

### 2.3. Shaping Process

The extruded strands were cut into pieces and compression-molded using a hot press PW40EH (Paul-Otto-Weber GmbH, Remshalden, Germany) with a compression molding time of 1 min at 260 °C. The plates were prepared with a diameter of 60 mm and a thickness of 0.5 mm.

### 2.4. Characterization

The SWCNT macrodispersion of the composites was investigated by transmission light microscopy using an integrated light microscope (Olympus BX 53M-RLA) equipped with an Olympus DP74 camera. The extruded strands were fixed by dipping them in a resin, and the cured resin with the embedded strand was cut using a Leica RM2265 instrument in thin slices with a thickness of 5 µm.

For characterisation of SWCNT nanodispersion, scanning electron microscopy (SEM) images of the composites were acquired using an ULTRA Plus (Carl Zeiss AG, Oberkochen, Germany) scanning electron microscope at 3 kV acceleration voltage using the SE2 detector. Cryo-fractured surfaces of strands were observed. All samples were sputtered with a 3-nm platin film.

The thermoelectric properties of strips (dimension 12 × 5 × 0.5 mm) cut from the compression molded samples were determined using the device developed and built at the Leibniz Institute of Polymer Research Dresden (IPF) [53]. The measurements were carried out at 40 °C and at the following temperature differences: 32–40 °C, 36–40 °C, 44–40 °C, and 48–40 °C. The Seebeck coefficients were determined by measurements of two to four strips. The individual values were obtained by repeating measurements five times for each temperature difference. Electrical volume resistivity measurements were carried out at 40 °C on the same equipment using the same samples. The 4-wire technique was used in the measurements and the mean values of the resistivity values were calculated with ten measurements. The Keithley multimeter DMM2001 (Keithley Instruments, Cleveland, OH, USA) was used for resistance and thermovoltage measurements. More details on these measurements are given in references [54,55]. The Seebeck coefficient S was calculated from the quotient of the thermoelectric voltage U and the applied temperature difference dT. The power factor PF is calculated from the electrical volume conductivity σ and the squared Seebeck coefficient (PF = S^2^·σ).

The electrical volume resistivity measurement of the compression-molded samples was performed using a Keithley 8009 Resistance test fixture coupled with a Keithley electrometer E6517A at room temperature. For samples with a resistance lower than 10^7^ Ohm, strips were cut from these plates, and in-plane electrical resistivity was measured using a Keithley multimeter Model DMM2001 and a 4-point test fixture developed by IPF. Both surfaces of two different samples were measured for each composite sample. The averages of these values were taken.

The thermal behavior of the samples was measured using a Perkin Elmer DSC 4000 differential scanning calorimeter (DSC) under nitrogen atmosphere. The samples weighted 9.5 mg were heated from 25 to 250 °C at a heating rate of 10 K/min. The glass transition temperature (Tg) data were obtained from the second heating scan.

## 3. Results and Discussion

### 3.1. Composite Morphology

In the preparation of SWCNT-based polymer composites by the melt mixing process, the shear stresses to which the SWCNT primary material was exposed during the mixing process is of critical importance. Therefore, the shear stresses need to be optimized to achieve the desired dispersion and distribution without sacrificing the structural integrity of the SWCNTs [56]. With the increase of rotation speed, shear stresses or mixing energy input increased. To examine the effect of rotation speed, composites containing 0.1, 0.25, and 1 wt% SWCNT were prepared by direct incorporation at rotation speeds varying between 50 and 250 rpm. 

The macrodispersion of 1 wt% SWCNTs in PMMA composites at the different rotation speeds was studied using transmission light microscopy (LM) on thin sections (Figure 1). The number and size of remaining agglomerates decreased with increasing rotation speed. The morphologically best SWCNT dispersion was obtained for the sample prepared at 250 rpm, indicated by the lowest number of agglomerates. It can be concluded that high shear stresses are favorable for good dispersion and homogeneous distribution of the SWCNTs.

Composites containing 5 wt% SWCNTs were prepared by the direct incorporation method at 250 rpm and 50 rpm and regarded as a masterbatch, which was diluted to 0.05 to 1 wt% SWCNTs at 50 or 250 rpm. Figure 2 compares light microscope images of PMMA/1 wt% SWCNTs composites using the masterbatch approach at different rotation speeds in both steps. It can be seen that using 50 rpm in both steps caused a poor SWCNT dispersion with many large agglomerates (Figure 2a). The SWCNT dispersion is similar to that achieved with direct SWCNT incorporation (Figure 1e). If at least one mixing step took place at 250 rpm, significantly fewer agglomerates are seen in the composites (Figure 2b–d). Thereby, whether the higher rotation speed was applied in the masterbatch step or in the dilution step plays a subordinate role in the SWCNT dispersion.

Figure 3 shows SEM images that can be used to assess SWCNT distribution at the nanoscale in the composites. It is remarkable that for the composite prepared by direct incorporation at 50 rpm (Figure 3a), the SWCNT are distributed significantly differently than in Figure 3b–d. Here, areas with SWCNT are visible and areas where no SWCNT can be seen (marked with dotted lines), indicating a very inhomogeneous SWCNT distribution. This observation correlates with the light microscopy image of this composite (Figure 1e), which shows very large agglomerates and thus a very inhomogeneous macroscopic SWCNT distribution. It can be concluded that when SWCNTs are directly incorporated into PMMA at the lowest rotation speed of 50 rpm, the SWCNTs are distributed very inhomogeneously in both the nanoscopic and macroscopic scales. For the composite prepared by direct incorporation at the highest speed of 250 rpm (Figure 3b) and the two composites produced by masterbatch dilutions (Figure 3c,d), uniform SWCNT distribution on a nanoscopic scale is seen in each case. The rotation speed seems to play a subordinate role in the masterbatch approach for the SWCNT distribution in nanoscale. However, larger and more numerous agglomerates were observed in the light microscopy images when only 50 rpm (Figure 2a) was used in the masterbatch approach instead of 250 rpm (Figure 2b).

### 3.2. Electrical Properties

The electrical resistivity vs. the SWCNT content of the composites prepared by direct incorporation at 250 rpm is presented in Figure 4. A distinct decrease compared to the value of pure PMMA occurs at the addition of 0.05 wt% SWCNTs indicating the electrical percolation threshold to be in the range of 0.05 to 0.075 wt%. This is the lowest electrical percolation threshold value reported so far in literature for melt-mixed PMMA/SWCNT composites. The decrease in the electrical resistivity of the composites continued until the addition of 1 wt% SWCNT. At contents higher than 1 wt % SWCNT, the values levelled off at around 10 Ohm∙cm.

To examine the effect of a shear stresses on the electrical resistivity of the composites, the rotation speed of the composites containing 0.1 wt%, 0.25 wt%, and 1 wt% SWCNTs was varied between 50 and 250 rpm. It appeared that the electrical resistivity decreased with the increasing rotation speed (Figure 5). The rotation speed effect is most pronounced at the filler content of 0.1 wt% SWCNTs. This can be explained by the fact that this is the concentration closest to the electrical percolation threshold, making the conductive network with the small number of contact points very sensitive to changes. Obtaining the lowest electrical resistivity value at the highest speed may be due to the high shear stresses making it easier to separate the SWCNT bundles. As seen in Figure 1, only a small amount of agglomerate is observed at high rotation speeds, whereas at 50 rpm, a high number of agglomerates are visible. These results of better dispersion and at the same time reduced resistivity at increasing rotation speed is what would be expected. However, this is in contrast to some other investigations on MWCNT composites. An example is the study by Krause et al. [31] on PA6/MWCNT (type NC7000) composites, where it was found that the percolation threshold increased from 2–2.5 wt% to 3–4 wt% when the rotation speed was increased from 50 to 150 rpm. A comparable trend was observed by Pötschke et al. [35] for composites of 0.5 wt% MWCNT NC7000 in poly(caprolactone) (PCL), where the electrical resistivity increased with increasing rotation speed (50–400 rpm) whereby at the same time the number of residual agglomerates in the polymer matrix decreased. The explanation of those effects, which on the first view are unexpected, was a possible MWCNT length shortening at higher rotation speeds [56]. However, in our case, for the SWCNTs investigated, high rotational speed of 250 rpm led to both a more homogeneous SWCNT dispersion and a lower resistivity, indicating that length shortening possibly may not have a significant effect.

The electrical resistivity results of the composites prepared by direct incorporation and masterbatch dilution are compared in Figure 6. The masterbatch (PMMA/5 wt% SWCNT) was prepared at 50 rpm or 250 rpm. Both masterbatches were diluted at 50 or 250 rpm in the dilution step.

The highest resistivity values were achieved at the rotation speed of 50 rpm in the masterbatch method and in the direct incorporation method. In particular, the electrical resistivity values of the composites are higher in composites prepared at a rotation speed of 50 rpm in both the masterbatch preparation and dilution step. The electrical resistivity of composites with 0.05 wt% and 0.075 SWCNT prepared at the masterbatch preparation speed of 250 rpm (regardless of high or low dilution speed) were three and four orders of magnitude higher, respectively, than those of composites prepared by direct incorporation at 250 rpm. Consequently, the electrical percolation threshold of the composites prepared with the masterbatch dilution method was higher compared to that using the direct incorporation. The best results (lowest resistivities) for almost all samples were obtained at a rotation speed of 250 rpm for both preparation approaches (MB and MBD). It was concluded that the rotation speed of 50 rpm was ineffective in masterbatch preparation and dilution as well as in direct incorporation. The electrical resistivities of the all composites prepared by the masterbatch dilution approach were up to a SWCNT content of 0.5 wt% higher than those prepared by the direct incorporation. At a SWCNTs content of 1 wt%, the resistivity values were in the same range and no significant influence of the preparation method can be observed. As a result, the most effective way to prepare composites with good dispersion and low electrical resistivity is the direct incorporation at 250 rpm.

### 3.3. Thermoelectrical Properties 

The results of the thermoelectric properties of PMMA/SWCNT composites measured at 40 °C are given in Figure 7 and Appendix A. The thermoelectric measurements were started with composites having a SWCNT concentration as low as 0.25 wt% and were carried out up to 5 wt%. All composites exhibit positive Seebeck coefficients due to the p-type characteristic of SWCNTs of the type Tuball™. The Seebeck coefficient, which increased with the addition of SWCNTs, achieved a maximum value of 53.4 µV/K at 2 wt% filler. 

The power factor of the composites was calculated based on the Seebeck coefficient and electrical volume conductivity. The highest power factor value was 0.207 µW/m·K^2^ at 3 wt% SWCNT loading. For PA6/PMMA/SWCNT composites (90/10/3), a maximum power factor of 0.135 µW/m·K^2^ was previously determined at the same amount of filler content [43].

Comparing the Seebeck coefficient values of the composites with those of the pure SWCNT powder with a value of 39.6 µV/K [44] indicates that the Seebeck coefficient value increases after incorporation these SWCNTs into the PMMA polymer Matrix. This implies that there is a p-doping effect of the PMMA chains on the SWCNTs. A similar effect has already been described for polyether ether ketone (PEEK)/SWCNT Tuball [38] and polybutylene terephthalate (PBT)/SWCNT Tuball composites [44,54].

### 3.4. Thermal Behavior 

The thermal behavior of PMMA/SWCNT composites was investigated by DSC analysis. The Tg value of pure PMMA, which was 113.7 °C, increased to 114.6 °C with the addition of 0.05 wt% SWCNTs. The Tg of the composites increased further with the addition of higher SWCNT amounts and reached 117.0 °C at a filler loading of 5 wt%. It is known that when an inorganic filler is incorporated into an organic polymer matrix, the chain mobility of polymers is affected by polymer–surface interactions [57]. Therefore, the increased Tg due to the addition of SWCNTs can be attributed to the fact that the mobility of PMMA chains is significantly reduced after mixing with SWCNTs. In other words, adding SWCNTs to the PMMA matrix reduced the free volume and chain mobility of PMMA. As a result, the Tg values increased with the increase of the amount of SWCNTs in the composites [22,57,58]. This increase is in agreement with other results. Badawi and Al Hosiny [58] found Tg values of 91.2, 92.7, and 99.5 °C for pure PMMA, 0.5 wt% SWCNTs, and 2 wt% SWCNT filled composites, respectively. In a study by Flory et al. [22], the Tg of pure PMMA increased from 99 °C to 116 °C with 1 wt% SWCNT loading.

The effect of the rotation speed and the masterbatch preparation method on the Tg of the preapared nanocomposites was also investigated. However, neither had a significant impact on the Tg of PMMA (Appendix A).

## 4. Conclusions

Electrically conductive PMMA/SWCNT nanocomposites were fabricated by melt mixing using direct incorporation and masterbatch dilution approaches. The volume resistivity of the composites varied between 1.81 × 10^6^ and 9.92 Ohm∙cm with the addition of 0.05–5 wt% SWCNTs to the insulating PMMA matrix. According to the volume resistivity results, the percolation threshold was in the range of 0.05–0.075 wt% SWCNTs. The investigation established that 250 rpm was the most effective rotation speed to get low electrical resistivity and good macrodispersion of the composites, regardless of whether the SWCNTs were incorporated directly or via the masterbatch approach. The number and size of agglomerations and the electrical resistivity decreased at higher rotation speeds. Masterbatches were prepared and diluted at low and high rotation speeds to examine the effect on electrical and morphological properties. Masterbatch samples prepared and/or diluted at 50 rpm were ineffective compared with those prepared by direct incorporation at 250 rpm in terms of dispersion and electrical properties. Similar results were obtained with samples prepared by direct incorporation at 250 rpm compared to samples prepared and diluted at 250 rpm (except for composites with low SWCNTs content). This result shows that, for the studied type of SWCNT, the selection of a suitable rotation speed is more important than the SWCNT addition method to produce good quality PMMA/SWCNT composites.

In the thermoelectric measurements on composites prepared by direct incorporation at 250 rpm, it was found that the Seebeck coefficient increased up to 2 wt% filler content and reached the maximum value of 53.4 µV/K. A high power factor, reaching values up to 0.207 µW/m·K^2^, was recorded in the composite filled 3 wt% SWCNTs. The electrical conductivity increased with the addition of SWCNTs, and the maximum value was obtained at 5 wt% with 145 S/m. 

Furthermore, DSC analysis showed that the addition of SWCNT increased the glass transition temperature of the PMMA by approximately 4 K.

In summary, the Tuball™ SWCNT material is an effective filler to obtain conductive composites with good dispersion and low electrical resistivity by melt-mixing. The thermoelectric measurements indicate that PMMA/SWCNT composites can be used as a thermoelectric material. 

## Figures and Tables

**Figure 1 nanomaterials-13-01431-f001:**
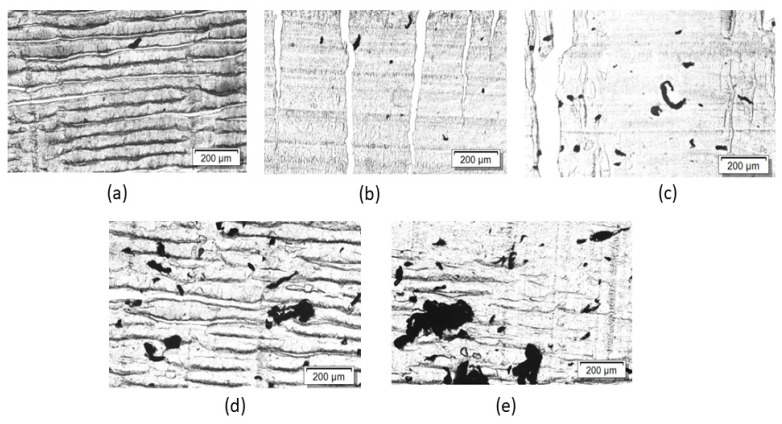
LM images of samples containing 1 wt% SWCNT prepared by direct incorporation at different rotation speeds: (**a**) 250 rpm, (**b**) 200 rpm, (**c**) 150 rpm, (**d**) 100 rpm, (**e**) 50 rpm (lines in the images are related to the brittleness of the samples and represents folds from the cutting process).

**Figure 2 nanomaterials-13-01431-f002:**
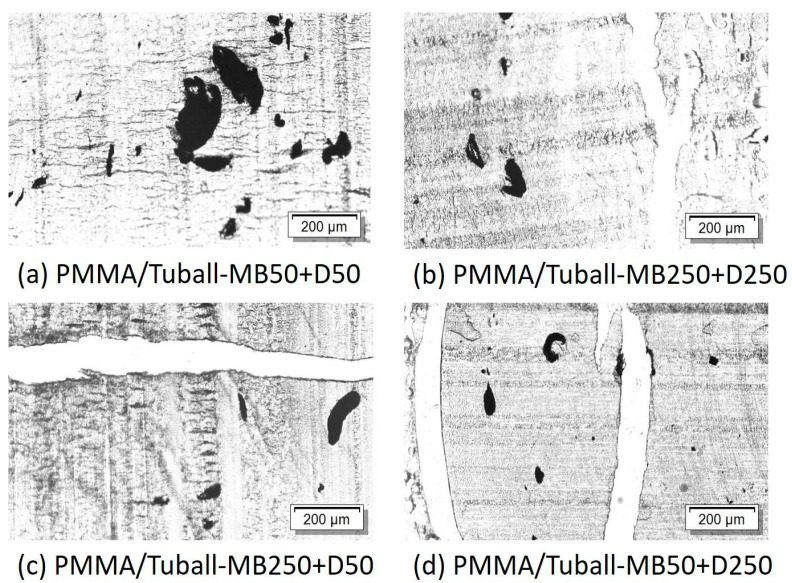
LM images of samples containing 1 wt% SWCNT prepared by masterbatch approach at different rotation speeds. 50 rpm/50 rpm (**a**), at 250 rpm/250 rpm (**b**), at 250 rpm/50 rpm (**c**), at 50 rpm/250 rpm (**d**) (rotation speed at masterbatch preparation/rotation speed at masterbatch dilution) (lines in the images are related to the brittleness of the samples and represents folds from the cutting process).

**Figure 3 nanomaterials-13-01431-f003:**
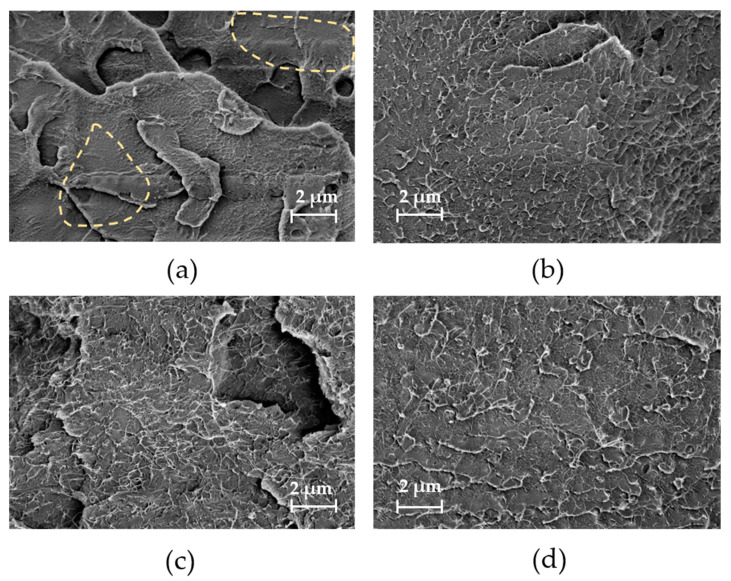
SEM images of cryofractures surface of composites containing 1 wt% SWCNT prepared by direct incorporation at different rotation speeds: (**a**) 50 rpm (dotted lines for highlighting), (**b**) 250 rpm, and masterbatch approach at different rotation speeds. 50 rpm/50 rpm (**c**), at 250 rpm/250 rpm (**d**) (rotation speed at masterbatch preparation/rotation speed at masterbatch dilution).

**Figure 4 nanomaterials-13-01431-f004:**
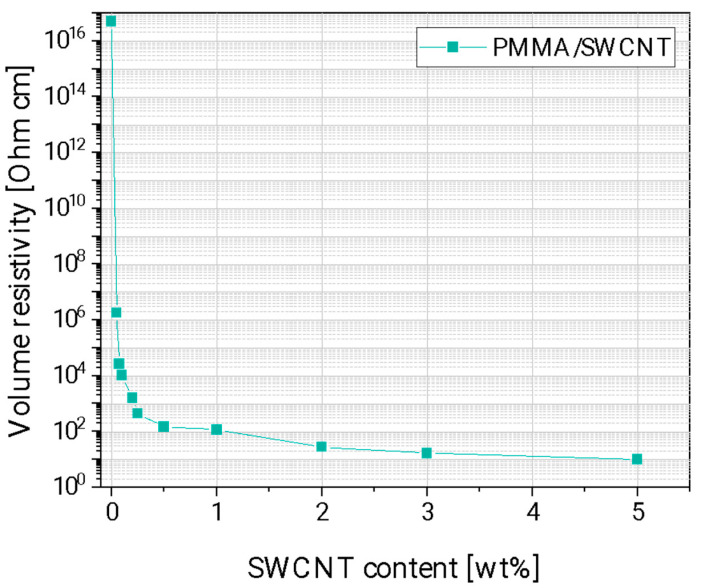
Electrical volume resistivity of PMMA/SWCNT composites prepared with direct incorporation method at 250 rpm.

**Figure 5 nanomaterials-13-01431-f005:**
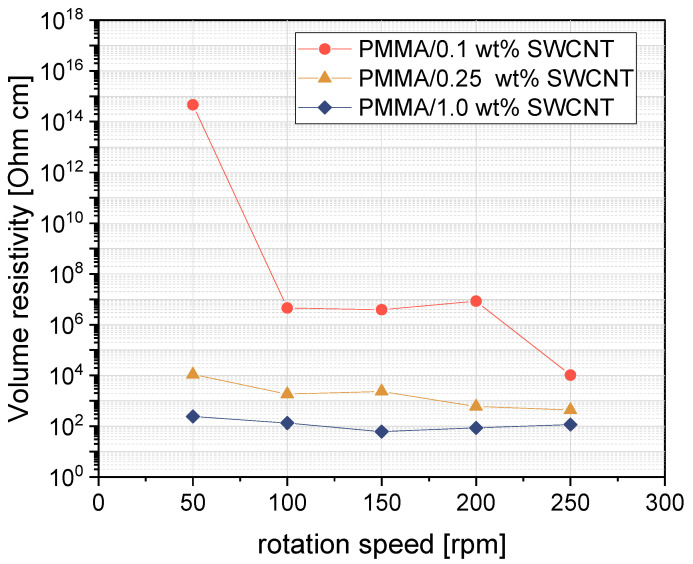
Electrical volume resistivity of PMMA/SWCNT composites prepared with the direct incorporation method at different rotation speeds.

**Figure 6 nanomaterials-13-01431-f006:**
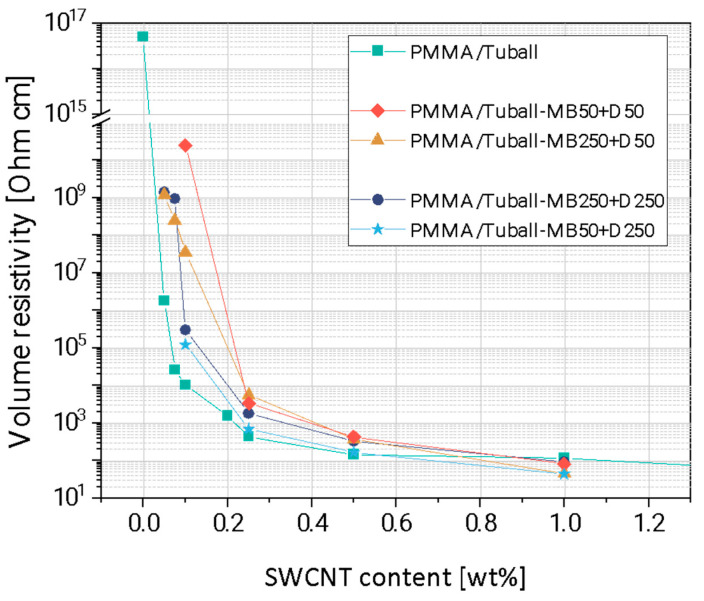
Electrical volume resistivity of PMMA/SWCNT composites prepared with the masterbatch diluting method at different rotation speeds compared with PMMA/SWCNT samples prepared at 250 rpm using direct incorporation.

**Figure 7 nanomaterials-13-01431-f007:**
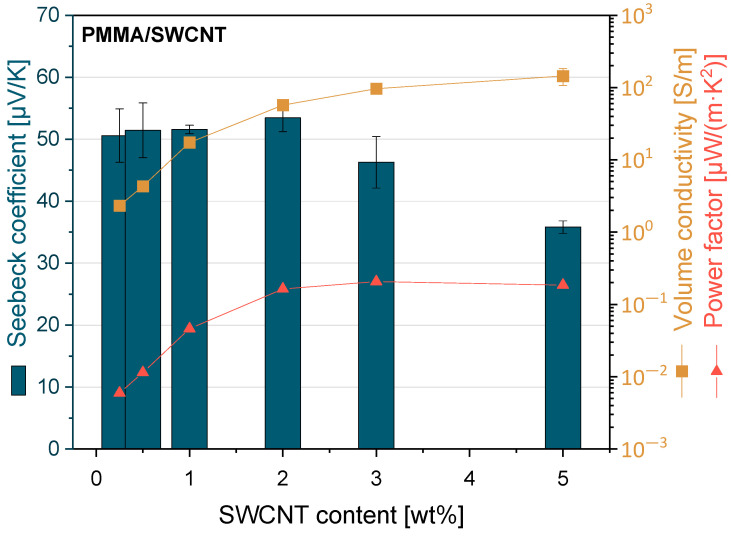
Thermoelectric properties of PMMA/SWCNT composites prepared by direct incorporation at a rotation speed of 250 rpm.

## Data Availability

The data presented in this study are available on request from the corresponding author.

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
