# Peer review of "PMMA/SWCNT Composites with Very Low Electrical Percolation Threshold by Direct Incorporation and Masterbatch Dilution and Characterization of Electrical and Thermoelectrical Properties"

_nanomaterials, 2023, doi:10.3390/nano13081431_

Round 1
Reviewer 1 Report
This work reported the preparation of PMMA/SWCNT composites by melt mixing to achieve suitable SWCNT dispersion and distribution and low electrical resistivity, whereby the SWCNT direct incorporation method was compared with masterbatch dilution. The reviewer thinks that it can be considered for potential publication after some revision.
1. Abstract section: Please add the full name of abbreviations such as PMMA and SWCNT for the first use.
2. Introduction section: The nanocomposites of polymer and carbon materials are widely used in many fields, and CNTs-based composites always have good properties. The following new references can be involved to optimize the Introduction section.
https://doi.org/10.1016/j.mtla.2020.100704
https://doi.org/10.1016/j.snb.2023.133309
https://doi.org/10.1016/j.cej.2021.133915
https://doi.org/10.1016/j.jmrt.2023.03.129
https://doi.org/10.1021/acsami.2c01086
https://doi.org/10.1016/j.foodchem.2020.128484
https://doi.org/10.3390/bios12050354
3. The purities of reagents and materials are important in Section of 2.1. Materials
4. Please revise the description of horizontal and vertical coordinates in Figs. 3-6
Author Response
Comments and Suggestions for Authors
This work reported the preparation of PMMA/SWCNT composites by melt mixing to achieve suitable SWCNT dispersion and distribution and low electrical resistivity, whereby the SWCNT direct incorporation method was compared with masterbatch dilution. The reviewer thinks that it can be considered for potential publication after some revision.
- Abstract section: Please add the full name of abbreviations such as PMMA and SWCNT for the first use.
Answer: We added the abbreviations in the abstract
- Introduction section: The nanocomposites of polymer and carbon materials are widely used in many fields, and CNTs-based composites always have good properties. The following new references can be involved to optimize the Introduction section.
https://doi.org/10.1016/j.mtla.2020.100704
https://doi.org/10.1016/j.snb.2023.133309
https://doi.org/10.1016/j.cej.2021.133915
https://doi.org/10.1016/j.jmrt.2023.03.129
https://doi.org/10.1021/acsami.2c01086
https://doi.org/10.1016/j.foodchem.2020.128484
https://doi.org/10.3390/bios12050354
Answer: Thank you for your suggestions. We have inserted the references.
- The purities of reagents and materials are important in Section of 2.1. Materials
Answer: We used the virgin PMMA and therefore we added the word “pure” before PMMA to indicate that we used 100% PMMA. For the used CNTs we re-wrote the sentence to be more clear: The carbon purity of SWCNTs is 75%.
- Please revise the description of horizontal and vertical coordinates in Figs. 3-6
Answer: Unfortunately, we do not understand the referee comment. All axes are labelled with the parameters including the unit. We changed nothing.
Reviewer 2 Report
The manuscript by Uçar et al. entitled:
"PMMA/SWCNT composites with very low electrical percolation threshold by direct incorporation and masterbatch dilution and characterization of electrical and thermoelectrical properties"
reports the study of the electrical (electrical resistivity), thermoelectrical (Seebeck coefficient) and thermal (glass transition) properties of PMMA/SWCNT composites prepared under different experimental melt mixing conditions. The very small electrical percolation threshold of 0.05-0.075 wt% was found, and 250 rpm was determined as the most effective rotation speed to get low electrical resistivity and good macrodispersion of the composites. A Seebeck coefficient higher than 50 mV/K was observed for SWCNT contents up to 2%, decreasing for higher concentrations. It is an interesting and manuscript, but contains some small mistakes that must be corrected before being accepted for publication in “Nanomaterials”.
In particular:
Sometimes spaces are missing between sentences (e.g. line 46, “thusa”, or line 113 “focuseddirect”); please check carefully and correct.
TEM observations of the SWCNT distribution could be included.
Figure 6 plots the Seebeck coefficient, volume conductivity and thermoelectric power factor as a function of SWCNT content. There is a significant decrease of the Seebeck coefficient for SWCNT contents higher than 2%, while the conductivity remains constant. However, the power factor remains roughly constant, which makes no sense. Please explain.
Author Response
The manuscript by Uçar et al. entitled:
"PMMA/SWCNT composites with very low electrical percolation threshold by direct incorporation and masterbatch dilution and characterization of electrical and thermoelectrical properties" reports the study of the electrical (electrical resistivity), thermoelectrical (Seebeck coefficient) and thermal (glass transition) properties of PMMA/SWCNT composites prepared under different experimental melt mixing conditions. The very small electrical percolation threshold of 0.05-0.075 wt% was found, and 250 rpm was determined as the most effective rotation speed to get low electrical resistivity and good macrodispersion of the composites. A Seebeck coefficient higher than 50 mV/K was observed for SWCNT contents up to 2%, decreasing for higher concentrations. It is an interesting and manuscript, but contains some small mistakes that must be corrected before being accepted for publication in “Nanomaterials”.
In particular:
Sometimes spaces are missing between sentences (e.g. line 46, “thusa”, or line 113 “focuseddirect”); please check carefully and correct.
Answer: We check the named sentences and corrected them.
TEM observations of the SWCNT distribution could be included.
Answer: We performed SEM study on cryofractured surface of four different composites to show the SWCNT distribution in the composite. We added some text in experimental part and added a Figure including description after the LM part.
Figure 6 plots the Seebeck coefficient, volume conductivity and thermoelectric power factor as a function of SWCNT content. There is a significant decrease of the Seebeck coefficient for SWCNT contents higher than 2%, while the conductivity remains constant. However, the power factor remains roughly constant, which makes no sense. Please explain.
Answer: When calculating the power factor (PF), the Seebeck coefficient (S) squared is used and the conductivity is a factor. Since the numerical value of the Seebeck coefficient is significantly lower (around 0.000050 V/K) than the numerical value of the conductivity (around 2-144 S/m), the value of the conductivity plays a greater role in the calculation of PF. Because of this mathematical relationship, the change in the S value has less effect on the PF value than the change in the conductivity.
For the 5wt% sample, the Seebeck coefficient decreases, but the conductivity increases strongly compared to the 2wt% sample. As a result, the PF values of the 2wt% and 5wt% composites are quite similar.
We have added the following table to the Supporting Information to see the values more clearly.
Table S1. Thermoelectric properties of PMMA/SWCNT composites
|
SWCNT Content wt% |
Seebeck coefficient µV/K |
Volume conductivity S/m |
Power factor µW/(m∙K2) |
|
0.25 |
50.6 ± 4.31 |
2.3 ± 0.0 |
0.006 |
|
0.5 |
51.4 ± 4.40 |
4.3 ± 0.4 |
0.011 |
|
1 |
51.6 ± 0.67 |
17.4 ± 2.8 |
0.046 |
|
2 |
53.4 ± 2.2 |
57.2 ± 9.6 |
0.163 |
|
3 |
46.3 ± 4.2 |
96.7 ± 4.2 |
0.207 |
|
5 |
35.8 ± 1.0 |
144.5 ± 38.0 |
0.185 |
Reviewer 3 Report
The use of PMMA/SWCNT composites as the thermoelectric is a hot research area and point at last 5 or ten year. But the first impression this work gives the reader is clearly not that of a complete scientific study and work, but only a report of a scientific experiment. Complete and corresponding experimental data, the specific conditions of the experiment and, most importantly, the scientific discussion and rationale are lacking.
1 The authors do not give specific conditions for the experiments, such as the test temperature, what the delt T is, and the measurement temperature range, which is the most important aspect of the thermoelectric properties of the Seebeck efficiency. Whether the data are linear or not.
2. LM images only show defects in the post-macroscopic material and cannot be observed at the scale of the carbon nanotube level. The authors must provide higher precision SEM, AFM or TEM images.
3. In Fig. 6, there is an increase in conductivity and Seebeck with increasing concentration of SWCNT doping, which is not reasonable in a thermoelectric material. The authors must give a reasonable explanation. Furthermore the units of power factor are not indicated in Fig. 6.
4. In comparison with other work, why the conductivity of this work is much better than the reported value, while the Seebeck coefficency is much lower than the reported value. https://pubs.rsc.org/en/content/articlelanding/2018/ta/c7ta11146k/unauth
Author Response
Comments and Suggestions for Authors
The use of PMMA/SWCNT composites as the thermoelectric is a hot research area and point at last 5 or ten year. But the first impression this work gives the reader is clearly not that of a complete scientific study and work, but only a report of a scientific experiment. Complete and corresponding experimental data, the specific conditions of the experiment and, most importantly, the scientific discussion and rationale are lacking.
1 The authors do not give specific conditions for the experiments, such as the test temperature, what the delt T is, and the measurement temperature range, which is the most important aspect of the thermoelectric properties of the Seebeck efficiency. Whether the data are linear or not.
Answer: Thank you for your comments. We have reported the conditions and explanations of thermoelectric measurements in more detail in Section 2.5.
- LM images only show defects in the post-macroscopic material and cannot be observed at the scale of the carbon nanotube level. The authors must provide higher precision SEM, AFM or TEM images.
Answer: We performed SEM study on cryofractured surface of four different composites to show the SWCNT distribution in the composite. We added some text in experimental part and added a Figure including description after the LM part.
- In Fig. 6, there is an increase in conductivity and Seebeck with increasing concentration of SWCNT doping, which is not reasonable in a thermoelectric material. . Furthermore the
Answer: For Polymer/SWCNT composites the dependence of Seebeck coefficient on the SWCNT content was studied for different polymer types. Please read https://www.mdpi.com/2504-477X/3/4/106 . Here, you can find results for PA6 and ABS composites. In both cases, a maximum S-values at 0,5-1 wt% SWCNT could be observed. For PBT/SWCNT an increase of S-value with SWCNT content was found. In contrast, for PEEK/SWCNT and PC/SWCNT composites, the S-values decrease with increasing SWCNT content (Konidakis, I.; Krause, B.; Park, G.-H.; Pulumati, N.; Reith, H.; Pötschke, P.; Stratakis, E. Probing the carrier dynamics of polymer composites with single and hybrid carbon nanotube fillers for improved thermoelectric performance. ACS Appl. Energy Mater. 2022, 5, 9770–9781). Therefore, the dependency of Seebeck coefficient with SWCNT content is strongly dependent on the kind of polymer and does not always have a rising course.
The unit in Fig. 6 are given.
- In comparison with other work, why the conductivity of this work is much better than the reported value, while the Seebeck coefficency is much lower than the reported value.
Answer: Thank you for pointing out this paper. The results of Wu et al. and our investigations cannot be compared. The composites were produced in very different ways. Wu et al. made the composites by polymerisation and cross-linking while we incorporated the SWCNTs into the polymer melt by melt mixing. Furthermore, a different SWCNT type was used, which has a very strong influence on the Seebeck coefficient of the composites. S-values of different kind of CNTs are reported in J. Compos. Sci. 2019, 3(4), 106; https://doi.org/10.3390/jcs3040106. A comparison of the same SWCNT type after incorporation in these two different ways would be very interesting and this would then be comparable (without Sn). In addition, Wu et al. added Sn electrolyte, which releases electrons through oxidation/reduction reaction, which strongly influences the conduction mechanism and cannot be compared to electrical conduction by SWCNT only.
Round 2
Reviewer 3 Report
This work requires further research and more detailed scientific analysis to be given by the authors in the future, and there are still many questions that need to be experimented and explained.